# Product Quality Measurement, Dynamic Changes, and the Belt and Road Initiative Distribution Characteristics: Evidence from Chinese Wooden Furniture Exports

**Lu Wan [1], Nannan Ban [2,\*], Yizhong Fu [1] and Luyao Yuan [1]**

[1]    School of Economics and Management, Beijing Forestry University, Beijing 100083, China; wanlu@bjfu.edu.cn (L.W.); fuyizhong@bjfu.edu.cn (Y.F.); lu_yao_yuan@163.com (L.Y.)

[2]    Surrey International Institute, Dongbei University of Finance and Economics, Dalian 116025, China

[\*]    Correspondence: bannannan@dufe.edu.cn

**Abstract:** As the most important forest product in the export of China, wooden furniture is facing increasingly fierce international competition and has a strong need for quality improvement. Based on the endogenous determination model of quality, this paper measures the quality of Chinese wooden furniture in exports from 1998 to 2017, by using product-level trade data of BACI CEPII. From the perspectives of the overall and sub-category quality, it examines the characteristics of dynamic changes in the product quality and its regional distribution of "the Belt and Road Initiative" countries. The results show that the quality of Chinese wooden furniture in exports is lower than that of wood-based panels and paper products. It remains stable after a slight increase from 2001 to 2005, but the quality level is always low. Among the sub-categories, wooden furniture not for kitchens, offices, or bedrooms has the lowest quality, while wooden office furniture has the highest one. The three dominant sub-categories that account for a high export share are all low in quality, while the small proportion sub-categories are all of higher quality, implying a strong imbalance. In particular, the quality of the main export products, upholstered wooden seats and wooden furniture not for kitchen, office, or bedroom use, has continued to decline, highlighting the plight of the quality growth of Chinese wooden furniture. For the BRI markets, the quality of Chinese wooden furniture exported to the region has declined slightly since 2012. However, different markets have shown different characteristics in the quality level and the direction of change. In terms of quality level, the qualities of wooden furniture exported to Malaysia, Israel, the United Arab Emirates, Vietnam, and the Philippines are relatively high. In terms of changing trends, the qualities of wooden furniture to Malaysia, Thailand, Indonesia, and Israel are showing a rising trend. In this case, accurately identifying the quality of different export categories of furniture products and their changing characteristics can help furniture enterprises make better production and operation decisions, promote the formation of a good business environment, and foster new comparative advantages and international competitiveness.

**Keywords:** wooden furniture; quality of export products; measurement; dynamic changes; BRI distribution characteristics

## 1. Introduction

In the past two decades, the total exports of Chinese wood forest products have continued to rise. In 2018, the export value reached US $50.21 billion, which was 21.48 times the export value in 1998. Its proportion in the world's total value of wood forest product exports rose from 1.29% in 1998 to 12.64% in 2018. China has become the world's largest exporter of wood forest products (The export value of Chinese wood forest products and its share in the world's forest product exports are calculated by the author based on data from the UN Comtrade). Wooden furniture, paper products, and wood-based panels account for the majority of China's wood forest products exports, while roundwood, sawnwood,

and wood pulp take a small proportion [1]. Wooden furniture trade accounts for the largest proportion of the total value of China's wood forest product exports. In the past ten years, the proportion remained between 45% and 55% (The proportion of wooden furniture export in China's total wood forest product exports is calculated by the author according to the data from the UN Comtrade). In addition, the export competitiveness of Chinese wooden furniture is the strongest, followed by wood-based panels, and the export competitiveness of other products is weak. However, in terms of international market share, the proportion of Chinese wooden furniture exports in the world's total wooden furniture exports has declined year after year since 2015, from 35.66% in 2015 to 32.26% in 2018 (The proportion of China's wooden furniture exports in the world's total wooden furniture exports is calculated by the author according to the data from the UN Comtrade), facing strong international competition pressure. Therefore, we will conduct an in-depth discussion on wooden furniture as a representative wood forest product exported from China. With the rising labor cost in China, the price competitive advantage of wooden furniture exports has gradually weakened.

According to the statistical conclusions of Global Furniture Industry Development Outlook (China National Furniture Association, 2020), the slowdown in world trade and investment has had a certain degree of impact on the furniture industry. China's furniture production has declined. In particular, from the perspective of target market distribution, unlike traditional furniture producing areas, China's furniture exports are more global, which is greatly affected by the changes in the international market. On the contrary, European furniture is mainly traded within the region, and the market for products outside the region is not very open. In addition, the Chinese furniture industry itself also has problems such as large but not strong, insufficient independent innovation, and limited R&D investment [2]. In 2018, the growth rate of accumulated losses of Chinese furniture enterprises has increased significantly, and the risks of enterprise benefits have increased (China Furniture Industry Development Report, 2019).

Moreover, the comparative advantage is affected by trade protectionism such as anti-globalization. The trade frictions and trade barriers from developed countries have greatly increased. For example, in the Sino-US trade friction in 2018, Chinese wooden furniture for bedroom and kitchen use, and wooden furniture for office use, were all included in the US tariff list. The increase in trade barriers from developed markets has further exacerbated the cost disadvantage of Chinese wooden furniture. It has pushed Chinese furniture enterprises to recognize the factors that have a deep impact on product export besides price, i.e., product quality. This drives people to set their sights on the "the Belt and Road", a hot region for trade growth.

As early as "the Twelfth Five-year Plan", China proposed to optimize its foreign trade structure and accelerate the cultivation of new advantages with quality, technology, brand, and service as its core competitiveness. In November 2019, the Central and State Council issued the "Guiding Opinions on Promoting the High-Quality Development of Trade", which has become a programmatic document for promoting the high-quality development of China's foreign trade under the new situation. The quality of export products has become one of the key concerns of the Chinese government.

In the academic field, the economic research on product quality has received extensive attention. Scholars have used different methods to measure the quality of export products, explored the relationship between product quality and economic growth, competitiveness, and exports, and studied the factors affecting quality changes [3–5]. However, the existing research on product quality was mainly concentrated in areas such as large-scale manufacturing, food processing industry, textile industry, etc. The research on the economic quality of forest products is rare and further supplements are needed. For Chinese wood furniture, in view of its long-term active performance in world trade, scholars continue to pay attention to its trade structure and international competitiveness. With the introduction of new theories and new perspectives of international economics research, the focus has expanded from market share and competitiveness index [6], price changes and policy impacts [7,8],

to trade potential estimation [9], trade intensive and extensive margins' measurement [10], trade duration decision [11], etc. However, in terms of quantifying the economic quality of wooden furniture, the measurement and analysis that integrate the latest theoretical development of international trade are relatively scarce.

In addition, due to the complexity and multidimensional characteristics of product quality, it is difficult to establish accurate measurement indicators, and scholars continue to explore this. In general, there are four measurement methods used in the study of trade product quality: technical complexity [12], product unit value [13], export quality index [14], and regression estimation [15]. In the previous literature, the unit product value was often used to measure the quality of export products. This study comprehensively considers the advantages and disadvantages of various product quality measurement methods and the need to reflect the quality characteristics of the latest period. Thus, the regression method is adopted as the quality measurement method, to accurately measure the quality of wooden furniture in China's exports. On this basis, the paper further examines its dynamic changes and distribution characteristics, in order to provide the latest and reliable evidence for in-depth analysis and description of the quality of Chinese wooden furniture products in exports.

Our paper enriched the research in several ways. First, the existing literature focused on the influencing factors of product quality, and there were not many discussions on the quality level of export products from developing countries. The analysis of the current quality of Chinese specific export products in our study is a supplement to this aspect. Secondly, China is a country with very large forest products trade, but there are still few studies on the quality of forest products exported. Our study will focus on the main forest products of wooden furniture, which will broaden the existing research content. In addition, with the increasing uncertainty of trade policies in developed markets, taking advantage of the trade vitality in emerging markets has become a very valuable issue. The quality of export products varies with different destinations. This study extends the discussion of wooden furniture quality to the "Belt and Road" area, which provides an important quality analysis basis for wooden furniture products to develop new markets and strengthen regional cooperation.

The rest of the paper is organized as follows: Section 2 measures the quality of Chinese wooden furniture export products and explores its dynamic changes. Section 3 analyzes the changes in the scale and competitiveness of wooden furniture export. Section 4 discusses the distribution characteristics of Chinese wooden furniture quality in the "Belt and Road" market. Section 5 presents the conclusions.

## 2. Methodology

### 2.1. Theoretical Framework for the Quality Measurement of Export Products

Product quality can be defined in a broad sense and a narrow sense. In a narrow sense, product quality usually refers to certain product characteristics that can be measured by product quality indicators, such as strength, hardness, and chemical composition. Fulfilling these product quality requirements is the basis for product circulation in the market. In a broad perspective, product quality also includes certain features that cannot be measured by product quality indicators, such as shape and color. Product quality in economics research is a broad sense of quality. Quality measurement is mainly to establish relevant indicators so that the indicators can comprehensively reflect the various characteristics of products that meet the consumers' needs.

Economists started late in the study of product quality. Aiginger [16] proposed that product quality is the characteristic that consumers are willing to pay for. Kuhn and Mcausland [17] pointed out that quality is the attraction of products to consumers. It can be seen that the economic measurement of quality starts from the consumers. By taking needs fulfillment as a condition, it measures the degree to which products meet the needs of customers. It is the relative quality of the product that customers actually feel. It is the expression of the actual experience of the product from the standpoint of the consumer after comparing it with the competitor. Moreover, it is the embodiment of the product competitive advantage beyond the price. Therefore, this paper adopts the regression

estimation to measure the quality of export products. The basic idea of the regression method is to estimate the product consumption demand function with the information of product price and quantity, and then deduce product quality information.

The level of consumer utility is affected by both quantity and quality. Referring to the theoretical model of Hallak and Sivadasan [15], the quality index of firm's export product is introduced into the utility function, assuming that the consumer's utility function is:

$$U_p^c = [\sum_p (\lambda_p q_p)^{\frac{\sigma-1}{\sigma}}]^{\frac{\sigma}{\sigma-1}} \tag{1}$$

where $U_p^c$ refers to the utility of consumers in importer country $c$ of product $p$, $\lambda_p$ refers to the quality of product $p$, $q_p$ is the quantity of product $p$, and $\sigma$ is the substitution elasticity of the product. The price index corresponding to this consumption function is:

$$p = \sum_p p_p^{1-\sigma} \lambda_p^{\sigma-1} \tag{2}$$

The consumer demand function corresponding to product $P$ is:

$$q_{ct} = p_{ct}^{-\sigma} \lambda_{ct}^{\sigma-1} \frac{E_{ct}}{P_{ct}} \tag{3}$$

where $p_{ct}$ refers to the price of the product exported to country $c$ in year $t$, $E_{ct}$ is the expenditure of consumer in country $c$ in year $t$, and $P_{ct}$ is the price index corresponding to the consumption function of country $c$ in year $t$. The equation showed that, in the vertically differentiated product market, the consumption of the product depends on both the quality of the product and the price of the product.

Then, we take the natural logarithm of Equation (3) and obtain the regression model as follows:

$$\ln q_{ct} = (\sigma - 1) \ln \lambda_{ct} - \sigma \ln p_{ct} + \ln E_{ct} - \ln P_{ct} \tag{4}$$

Letting $\chi_{ct} = \ln E_{ct} - \ln P_{ct}$, $\varepsilon_{ct} = (\sigma - 1) \ln \lambda_{ct}$, the above Equation (4) could be transformed into:

$$\ln q_{ct} = \varepsilon_{ct} - \sigma \ln p_{ct} + \chi_{ct} \tag{5}$$

where $\chi_{ct} = \ln E_{ct} - \ln P_{ct}$ is a country-time control variable that changed with time and country, reflecting the actual purchasing power of wooden furniture in country $c$ within year $t$. In this paper, GDP of the export destination is selected as the importing country-time control variable. $\varepsilon_{ct}$ is the quality of the products exported to country $c$ in year $t$, and is treated as residual.

Thus, we defined the quality as:

$$quality_{ct} = \ln \hat{\lambda}_{ct} = \frac{\hat{\varepsilon}_{ct}}{(\sigma - 1)} = \frac{\ln q_{ct} - \ln \hat{q}_{ct}}{(\sigma - 1)} \tag{6}$$

Equation (6) can be used to measure the quality of a certain HS code product exported from China to country $c$ in year $t$. Furthermore, in order to obtain the overall quality and conduct comparative analysis, we standardize the quality index of (5) and have the standardized quality indicators of a certain HS coded product exported from China to country $c$ in year $t$:

$$r - quality_{ct} = \frac{quality_{ct} - \min quality_{ct}}{\max quality_{ct} - \min quality_{ct}} \tag{7}$$

where $\min quality_{ct}$ and $\max quality_{ct}$ refer to the minimum and maximum values of a certain HS code product in all years and to all destination countries, respectively. The above-mentioned standardized quality indicators do not have measurement units and can

be compared at product and destination level according to the proportion of export trade value. Thus, the overall quality can be expressed as:

$$TQ = \frac{v_{mt}}{\sum\limits_{mt \in \Omega} v_{mt}} \times r - quality_{mt} \tag{8}$$

where $\Omega$ in Equation (8) is a collection of samples at a certain level, TQ is the overall quality, and $v_{mt}$ is the trade scale. The index is between 0 and 1; the larger the value, the better the quality.

In short, the measurement process is as follows. Using the CEPII BACI database, we extract the annual export value and quantity data of each HS code product, and calculate the export price according to the equation that the export price equals the ratio of the export value to the export quantity, then brings the price equation and quantity data of each HS code product into Equation (5) for regression, and calculates the quality value of each product according to Equation (6). Finally, we obtain the quality values of different product levels in different periods based on Equations (7) and (8).

### 2.2. Data and Sample Selection

The data consist of a panel of bilateral trade flows and country-level variables for all the countries listed in the CEPII BACI database. The key data required in product quality estimation is bilateral trade values and quantities, disaggregated at the sectoral level. Specifically, we define the sectors at an HS 6-digit lever. In this paper, we focus on the export of Chinese wooden furniture. In response to the classification in Section 2, the products of wooden furniture contain six HS 6-digit sub-categories. In order to better study the dynamic changes of quality over time, we try to select a longer time period, especially the interval that covers important events before and after China's accession to the WTO, before and after the financial crisis. Meanwhile, for the purpose of reflecting the latest characteristics of the quality of Chinese wood in exports, we choose the period from 1998 to 2017. The export value of HS 6-digit wooden furniture and its corresponding export quantity to various destinations come from the CEPII BACI database. The sample data of this study include three dimensions, i.e., time, export destinations, and products. In the selection of sample trading partners, we selected all the trading partners that China exported wooden furniture each year, so as to accurately measure the quality and capture the characteristics of as many markets as possible. The GDP data of each trading partner come from the World Bank database (World Development Indicators).

## 3. Changes in the Export Scale and Competitiveness of Chinese Wooden Furniture

### 3.1. Wooden Furniture Export Accounts for the Largest Share of Wood Forest Product Exports

There are differences in the definition of wood forest products between Chinese research institutions and international organizations(see Table 1).

**Table 1.** Classification of wood forest products.

| FAO | China Forestry Development Report | China Forestry Statistical Yearbook |
|---|---|---|
| roundwood | roundwood | industrial roundwood and other raw wood |
| wood charcoal, chips and particles, residues, pellets and other agglomerates | sawnwood | sawnwood |
| sawnwood and veneer sheets | wood-based panels and veneer sheets | wood-based panels |
| wood-based panels | wood products | wood products |
| pulp and recycled paper | paper products | wood pulp |
| paper and cardboard | wooden furniture | paper products |
| | wood chips | wooden furniture |
| | others (firewood, charcoal, etc.) | |

Source: FAO& SFGAC website.

The Food and Agriculture Organization of the United Nations (FAO) broadly divides forest products into six categories, including ① roundwood (wood fuel, industrial roundwood, etc.), ② wood charcoal, chips and particles, residues, pellets, and other agglomerates, ③ sawnwood and veneer sheets, ④ wood-based panels (including plywood, particleboard, oriented strand board, etc.), ⑤ pulp and recycled paper, ⑥ paper and cardboard. However, wood products and wooden furniture are not included in the statistics. China Forestry Development Report (by State Forestry and Grassland Administration of China) classifies wood forest products into eight categories, specifically ① roundwood, ② sawnwood (including sawnwood and special-shaped sawlogs), ③ wood-based panels and veneer sheets (including veneer sheets, plywood, particleboard, fiberboard and laminated wood), ④ wood products, ⑤ paper products (including wood pulp, paper and cardboard, paper or cardboard products, waste paper and waste paper pulp, printed matter, etc.), ⑥ wooden furniture, ⑦ wood chips, and ⑧ others (firewood, charcoal, etc.). The China Forestry Statistical Yearbook (by State Forestry and Grassland Administration of China) identifies wood forest products from seven categories, including industrial roundwood and other raw wood, sawnwood, wood-based panels, wood products, wood pulp, paper products, and wooden furniture. Considering and combining the above classifications, the wood forest products analyzed in this paper are classified as Table 2:

**Table 2.** Classification of wood forest products in this paper.

| Category | HS Commodity Code | HS Commodity Name |
|---|---|---|
| Roundwood | 4403 | Wood in the rough, whether or not stripped of bark or sapwood, or roughly squared |
| Sawnwood | 4407 | Wood sawn or chipped lengthwise, sliced peeled |
| Wood-based panels | 4408 | Veneer sheets and sheets for plywood and other wood sawn lengthwise, sliced peeled |
|  | 4410 | Particle board, oriented strand board (OSB) and similar board of wood or other ligneous materials |
|  | 4411 | Fibreboard of wood or other ligneous materials |
|  | 4412 | Plywood, veneered panels and similar laminated wood |
| Wood pulp | 47 | Pulp of wood or other fibrous cellulosic material; waste and scrap of paper or paperboard |
| Paper products | 48 | Paper and paperboard; articles of paper pulp, of paper or paperboard |
| Wooden furniture | 940161 | Seats; with wooden frames, upholstered |
|  | 940169 | Seats; wooden frames, not upholstered |
|  | 940330 | Furniture; wooden, for office use |
|  | 940340 | Furniture; wooden, for kitchen use |
|  | 940350 | Furniture; wooden, for bedroom use |
|  | 940360 | Furniture; wooden, other than for office, kitchen or bedroom use |

Source: FAO& SFGAC website.

Chinese wood forest products exports are mainly wooden furniture, paper products and wood-based panels, while roundwood, sawnwood, and wood pulp account for a very small proportion. Specifically (see Figure 1), from 1998 to 2018, the export of wooden furniture took the largest proportion of the total export of wood forest products. The proportion changed between 45% and 55% but showed a trend of first rising and then falling. Furthermore, the top proportion was achieved in 2013 at 54.30%, and the average annual proportion amounted to 48.45%. Looking into the export value, it rose from US $1.09 billion in 1998 to US $22.96 billion in 2018. However, it declined slightly in 2016 and then continued to rise slowly. For paper products, the export took second place in the total export of wood

forest products. Its share was kept between 25% and 45% but experienced a decline first and then an increase. The highest share was 42.54% in 1998, and the lowest was 28.96% in 2004, with an annual average of 34.87%. In reference to the export value of paper products, it presented an upward trend, rising from US $0.99 billion in 1998 to US $19.46 billion in 2018. With regard to wood-based panels, it ranked third in the total exports of wood forest products, with the proportion between 5% and 25%. The proportion went up first and then decreased and the highest was in 2007, accounting for 21.13%. The annual average proportion was 14.18%. In addition, the export of sawnwood ranked fourth, with its proportion in the total export of wood forest products declining year by year. In particular, the proportion dropped from 4.91% in 1998 to 0.36% in 2018, having an average annual proportion of 2.11%. Meanwhile, the export value of sawnwood had a first upward and then downward trend, changing from US $114.3 million in 1998 to US $178.5 million in 2018. The export in 2008 was the highest, amounting to US $401.4 million. Although the exports of wood pulp and roundwood were on the rise, the value was quite small, accounting for a very low proportion of the total export trade of wood forest products. The proportions in each year are less than 1%, and the average annual proportions are 0.31% and 0.078%, respectively. Specifically, the export value of wood pulp increased from US $9.9 million in 1998 to US $131.3 million in 2018, while the export of roundwood was the smallest and always remained at a low level compared with various wood forest products. In 1998, it was only US $125 million, which increased to USD 23.6 million in 2018.

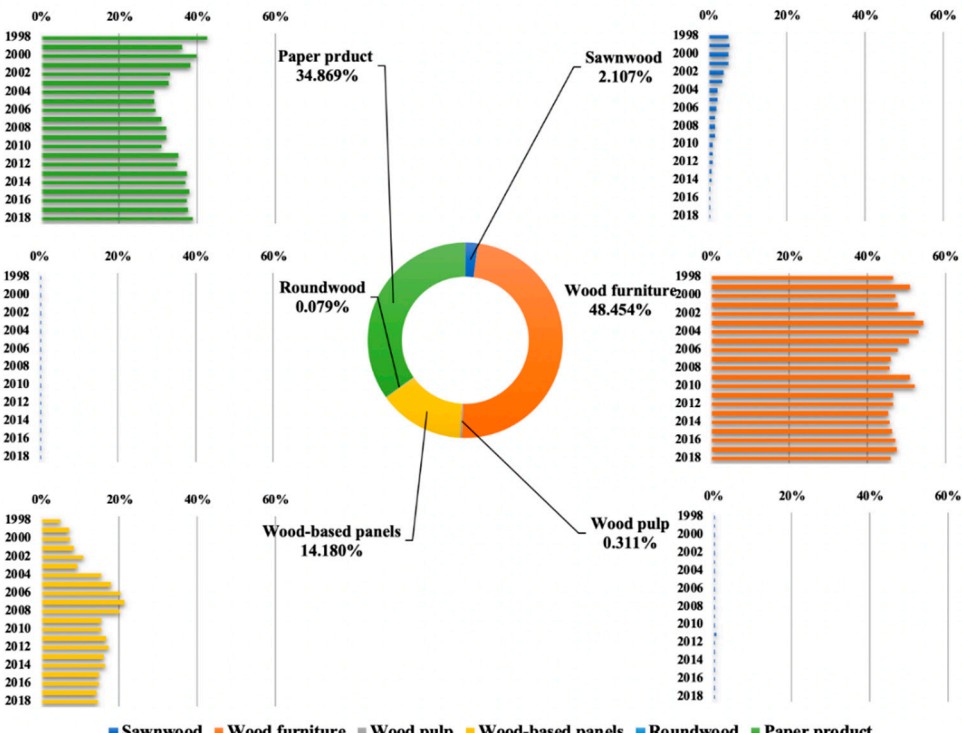

**Figure 1.** The proportion of the export value of wood forest products of China. Source: calculated according to the data of the United Nations UNCOMTRADE database (the percentage of the pie chart in the middle of Figure 1 represents an average export percentage between 1998 to 2018).

### 3.2. The Export Competitiveness of Wooden Furniture Is Significantly Stronger than That of Other Wood Forest Products

In comparison, among different wood forest products, the export competitiveness of wooden furniture is the strongest, followed by wood-based panels, according to the result in Figure 2. However, in recent years, the export competitiveness of wooden furniture has experienced a decline, while the export competitiveness of other products is weak to a certain extent. This section makes use of the revealed advantage index (RCA index) to

measure the export competitiveness of wood forest products. As shown in Figure 2, compared with other wood forest products of China, wooden furniture is the most competitive product. Excluding the great changes caused by the 2008 financial crisis, the competitiveness of wooden furniture continued to increase from 1998 to 2009, and the RCA index was between 1.25 and 2.5, which means the competitiveness of wooden furniture was strong and maintained growth. From 2010 to 2018, the RCA index exhibited a downward trend, with its value dropping from 2.88 to 2.43. In particular, the RCA index was greater than 2.5 in each year from 2010 to 2017, indicating a quite strong competitiveness. In addition, the export competitiveness of wood-based panels was second. From 1998 to 2006 and in 2008, the RCA index was less than 1.25, which means the export competitiveness was moderate. However, from 2006 to 2017, the RCA index was in the range of 1.25 to 2.5, which indicates that the export competitiveness of wood-based panels became strong. Although the competitiveness of the above products experienced improvement before 2007, they all encountered a downward turning point after 2010 (see Figure 2). This is because wooden furniture and wood-based panels are labor-intensive products. At the beginning of participating in international trade, with the advantage of low labor cost, the international market was rapidly expanded, and the competitiveness was continuously enhanced. In recent years, China's labor cost has increased year by year, the labor cost advantage has weakened, and export competitiveness has begun to show a downward trend.

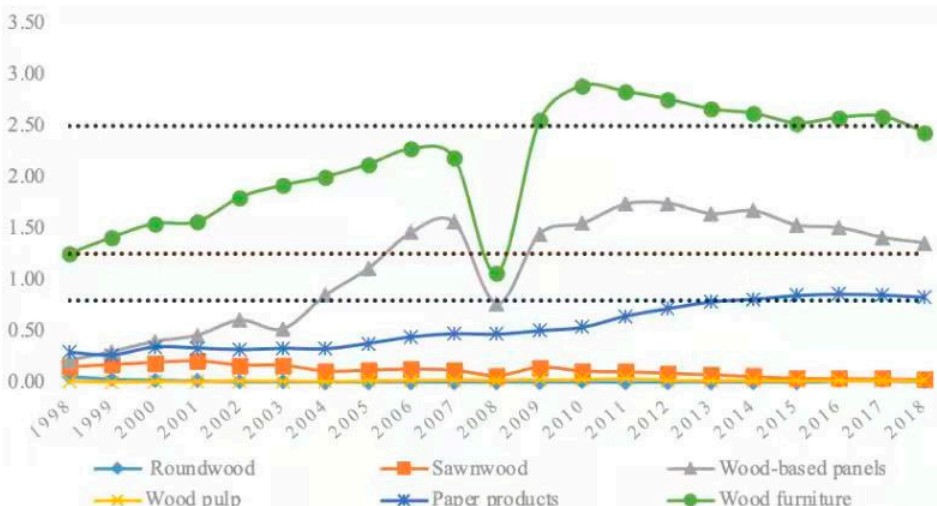

**Figure 2.** Changes in export competitiveness of wood forest products. Note: $RCA_{ij} = (X_{ij}/X_{tj}) \div (X_{iW}/X_{tW})$; $X_{ij}$ indicates the exporting value of goods I from country j; $X_{tj}$ indicates total exporting value of country j; $X_{iW}$ indicates the whole exporting value of goods I in the world; $X_{tW}$ indicates the total exporting value of the world. RCA > 2.5 means very strong competitiveness; 1.25 < RCA < 2.5 means strong competitiveness; 0.8 < RCA < 1.25 means moderate competitiveness; 0 < RCA < 0.8 means weak competitiveness (Balassa Bela, 1965). Source: calculated according to the data of the United Nations UNCOMTRADE database.

In addition, for paper products, although the export competitiveness showed an increase from 1998 to 2018, the competitiveness was always weak. The RCA index from 1998 to 2014 was less than 0.8, which was in a quite weak competitive position. From 2014 to 2018, the RCA index becomes slightly higher than 0.8, and the competitiveness is less weak. For other wood forest products, from 1998 to 2018, the RCA indexes of sawnwood, roundwood, and wood pulp were all below 0.8 in each year. Those export products were always in a very weak position in international competition, and the export competitiveness of sawnwood showed a clear decrease.

*3.3. The Export Competitiveness of Chinese Wooden Furniture Is Weaker Than That of Major European Competitors*

Although wooden furniture is the largest export category of wood forest products in China, there is still a significant gap between its export competitiveness and other traditional wooden furniture export powers. Using the global wooden furniture export data in 2018 to analyze the top ten export countries by trade value, as shown in Figure 3, we can see that, among the six sub-categories of wooden furniture, China's export value ranked first in four categories (940161, 940330, 940350, 940360) and ranked second in the other two categories (940169, 940340). Moreover, according to the statistics of the world wooden furniture export, 940360, 940161, 940350, and 940330 were the top four products in the world wooden furniture export, totaling 84.84%. Therefore, from the perspective of export scale, China plays an important role as a major supplier in the world wooden furniture market. To be sure, the export of wooden furniture is not only vital to the development of wood forest products trade in China, but at the same time, China, as a big exporter of such products, is also important to the sound development of the world wooden furniture market. However, since 2015, the export growth of Chinese wooden furniture began to significantly lag behind the growth of the world wooden furniture export, entering a period of transformation and development. More than 90% of Chinese furniture manufacturers are small and medium-sized enterprises. If these small and medium-sized enterprises rely on the competitiveness other than the price advantages, it is difficult to compete with established European and American furniture enterprises, not to mention the rapidly increasing trade protections in the North American and European markets. Therefore, certain emerging markets have gradually become important anchors for Chinese furniture enterprises to seek sustainable development.

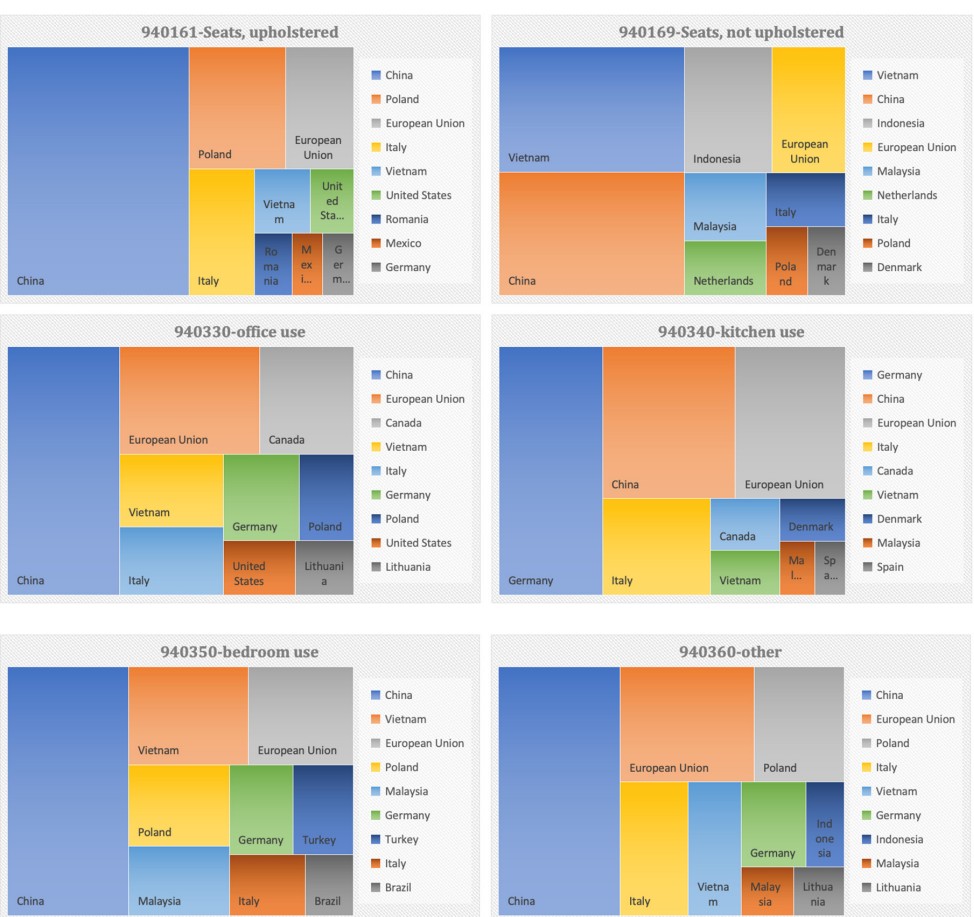

**Figure 3.** The world's major exporters of wooden furniture in 2018. Source: calculated according to the data of the United Nations UNCOMTRADE database.

Based on the analysis of the main exporters of the six sub-categories of wooden furniture, five developed countries and five developing countries ranked top were selected to calculate the revealed advantage index (RCA index) from 1998 to 2018, listed in Table 3. On this basis, we can further explore the characteristics in the change of export competitiveness of Chinese wooden furniture. Considering the meaning that the bigger the RCA, the stronger the competitive advantage, compared with these major exporting countries, it can be found that European countries, such as Poland, Denmark, and Italy, had very strong competitive advantages in wooden furniture exports (RCA was higher than 2.5 in each year), which was much higher than China. In the Asia-Pacific region, except that the export competitiveness of Vietnamese wooden furniture was significantly stronger than that of China, Chinese wooden furniture exports still had a strong advantage, in comparison with Malaysia, Indonesia, and other developing countries. Furthermore, this advantage was stronger than that of the developed countries in North America. However, when paying attention to the changes of China's RCA index over time, we can find that the competitive advantage began to enter a falling zone in 2015. In 2018, the RCA index fell to 2.44, which was less than 2.5, indicating that it had entered a tough development period with a relatively basic level of competitiveness.

**Table 3.** RCA Index of the world's major wooden furniture exporters.

| Year | Canada | China | Denmark | Germany | Indonesia | Italy | Malaysia | Poland | USA | Viet Nam |
|---|---|---|---|---|---|---|---|---|---|---|
| 1998 | •1.65 | •1.25 | •6.93 | •0.87 | •0.79 | •4.53 | •2.40 | •10.59 | •0.27 | - |
| 1999 | •1.72 | •1.41 | •6.26 | •0.89 | •2.75 | •4.44 | •2.55 | •10.46 | •0.25 | - |
| 2000 | •1.91 | •1.54 | •6.56 | •0.89 | •2.89 | •4.72 | •2.73 | •10.96 | •0.27 | •2.29 |
| 2001 | •1.93 | •1.56 | •6.16 | •0.88 | •2.96 | •4.53 | •2.54 | •10.17 | •0.27 | •2.69 |
| 2002 | •1.95 | •1.80 | •5.77 | •0.82 | •3.07 | •4.34 | •2.55 | •9.54 | •0.26 | •3.43 |
| 2003 | •1.90 | •1.92 | •5.90 | •0.76 | •3.04 | •4.17 | •2.58 | •9.35 | •0.26 | •5.10 |
| 2004 | •1.75 | •2.00 | •5.86 | •0.74 | •3.04 | •4.09 | •2.56 | •8.59 | •0.27 | •7.40 |
| 2005 | •1.60 | •2.12 | •5.19 | •0.86 | •3.18 | •3.82 | •2.56 | •8.12 | •0.29 | •8.27 |
| 2006 | •1.56 | •2.27 | •4.94 | •0.92 | •2.96 | •3.74 | •2.72 | •7.41 | •0.31 | •9.37 |
| 2007 | •1.25 | •2.18 | •4.66 | •0.91 | •2.64 | •3.60 | •2.65 | •6.99 | •0.32 | •9.45 |
| 2008 | •1.03 | •2.12 | •4.08 | •1.05 | •2.44 | •3.82 | •2.87 | •6.93 | •0.36 | •9.39 |
| 2009 | •0.89 | •2.55 | •3.67 | •1.08 | •2.24 | •3.40 | •2.87 | •6.50 | >•0.33 | •8.25 |
| 2010 | •0.86 | •2.88 | •3.64 | •1.03 | •2.15 | •3.34 | •2.87 | •6.78 | •0.35 | •9.08 |
| 2011 | •0.86 | •2.83 | •3.68 | •1.11 | •1.39 | •3.40 | •2.80 | •7.26 | •0.37 | •7.78 |
| 2012 | •0.86 | •2.75 | •3.64 | •1.05 | •1.80 | •3.31 | •2.86 | •6.72 | •0.36 | •7.44 |
| 2013 | •0.87 | •2.66 | •3.56 | •1.00 | •1.98 | •3.35 | •2.44 | •6.65 | •0.36 | •7.02 |
| 2014 | •0.80 | •2.62 | •3.53 | •0.93 | •2.01 | •3.16 | •2.30 | •6.55 | •0.32 | •6.58 |
| 2015 | •0.93 | •2.53 | •3.41 | •0.85 | •2.24 | •2.86 | •2.34 | •5.64 | •0.31 | •5.83 |
| 2016 | •1.04 | •2.58 | •3.43 | •0.86 | •2.14 | •2.74 | •2.35 | •5.88 | •0.28 | •5.55 |
| 2017 | •1.02 | •2.59 | •3.19 | •0.85 | •1.98 | •2.86 | •2.23 | •5.91 | •0.28 | •5.43 |
| 2018 | •0.97 | •2.44 | •3.12 | •0.86 | •1.97 | •2.74 | •2.07 | •6.14 | •0.27 | •5.34 |

Note: • refers to RCA > 2.5, i.e., very strong competitiveness; • refers to 1.25 < RCA < 2.5, i.e., strong competitiveness; • refers to 0.8 < RCA < 1.25, i.e., moderate competitiveness; • refers to 0 < RCA < 0.8, i.e., weak competitiveness. Source: calculated according to the data of the United Nations UNCOMTRADE database.

## 4. Discussion

### 4.1. Analysis on the Results of Quality Measurement of Chinese Wooden Furniture in Exports

In general, the quality of wooden furniture in exports is lower than that of wood-based panels and paper products. The quality value of wooden furniture in exports is between 0.2 and 0.25. From 1998 to 2001, it declined slightly but experienced a small increase during 2001 to 2005, the quality level of 2001–2005. After that, the quality level remained stable; nevertheless, it was always in a low position. As mentioned above, the quality measurement results can be compared at a different product level or at different destination level. Therefore, in the existing literature on quality measurement, the estimated quality values are not given direct attribute meanings. The economic significance is more reflected in the horizontal comparison of different products, different regions, and the vertical

comparison of different years. Here we describe the quality value of wooden furniture in exports as "low", for the following reasons. On the one hand, compared with other wood forest products, according to our estimations in other studies, the quality of Chinese paper products exported is higher than 0.35 in each year from 1998 to 2017 [18], which can even reach more than 0.5 for many sub-categories. On the other hand, if compared with the overall level of manufacturing industry, according to Shi (2013),the average overall quality of China's manufacturing export products is 0.821, which can also be concluded that the quality value of Chinese wooden furniture in exports is still in a low position.

Furthermore, we calculated the standard deviation of quality for major disaggregated wood forest products exported in each year, and the results are listed in Table 4. The standard deviations of qualities of the three products all showed a U-shaped change trend that first decreased and then increased, indicating that the product quality difference within the industry first narrowed and then expanded. After the new products entered the market, the enterprises within the industry followed suit, then the product quality gaps gradually narrowed, and the profits were reduced. Subsequently, in order to maintain their competitive advantages, high-productivity enterprises further developed and innovated, which produced high-quality new products. The product quality differences within the industry gradually expanded in line with the product life cycle. Each product is currently in a new round of innovative development and the establishment of new competitive advantages.

**Table 4.** Standard deviation of quality of Chinese major wood forest products.

| | 1998 | 2000 | 2002 | 2004 | 2006 | 2008 | 2010 | 2012 | 2014 | 2016 | 2017 |
|---|---|---|---|---|---|---|---|---|---|---|---|
| Wooden furniture | 0.151 | 0.135 | 0.132 | 0.131 | 0.135 | 0.138 | 0.140 | 0.145 | 0.140 | 0.142 | 0.142 |
| Wood-based panels | 0.159 | 0.147 | 0.131 | 0.128 | 0.136 | 0.115 | 0.093 | 0.108 | 0.113 | 0.112 | 0.114 |
| Paper products | 0.132 | 0.134 | 0.126 | 0.124 | 0.139 | 0.111 | 0.110 | 0.120 | 0.122 | 0.126 | 0.134 |

Specifically, in terms of the quality level of each sub-category of export wooden furniture (see Table 5), furniture not for kitchens, offices, or bedrooms under code 940360 accounted for the highest proportion of export in average, reaching 44.34%. However, its quality level was the lowest, with an average quality value of 0.184. The furniture for office use under code 940330 had the highest quality, with an average quality value of 0.702, but its export share was only 4.26%, a relatively low proportion of the total wooden furniture export. In addition, the qualities of other wooden furniture products were not high, having a quality value between 0.2 and 0.3.

**Table 5.** Quality of wooden furniture by sub-category.

| Category | HS Commodity Code | Average Quality | Annual Average Trade Proportion | Quality Change |
|---|---|---|---|---|
| Wooden furniture | 940161 | 0.245 | 26.52% |  |
| | 940169 | 0.267 | 6.35% |  |
| | 940330 | 0.702 | 4.26% |  |
| | 940340 | 0.244 | 4.14% |  |
| | 940350 | 0.219 | 14.41% |  |
| | 940360 | 0.184 | 44.34% |  |

Furthermore, we analyze the quality of Chinese wooden furniture in each sub-category exported in from 1998–2017. First of all, from Table 6, we can conclude that, among the six sub-categories of Chinese wooden furniture, products with large export proportions were 940161, 940360, and 940350, while the other three sub-categories all accounted for

less than 10%. In particular, the export proportion of upholstered wooden seats (940161) increased significantly over years, while other wooden furniture not for kitchens, offices, or bedrooms (940360) fell sharply year after year. In addition, the share of bedroom wooden furniture (940350) showed a characteristic of first increasing and then decreasing.

**Table 6.** Export value share of wooden furniture products by HS 6-digit category.

| Year | 940161 | 940169 | 940330 | 940340 | 940350 | 940360 |
|------|--------|--------|--------|--------|--------|--------|
| 1998 | 9.00% | 11.20% | 4.70% | 3.00% | 9.90% | 62.20% |
| 1999 | 9.80% | 10.30% | 4.20% | 2.80% | 12.20% | 60.70% |
| 2000 | 10.00% | 9.10% | 3.90% | 2.80% | 16.40% | 57.70% |
| 2001 | 11.30% | 8.40% | 2.50% | 3.20% | 18.80% | 55.80% |
| 2002 | 12.70% | 7.70% | 3.10% | 3.20% | 20.70% | 52.70% |
| 2003 | 17.30% | 7.90% | 3.80% | 3.50% | 13.30% | 54.10% |
| 2004 | 22.90% | 5.50% | 3.80% | 3.40% | 13.10% | 51.30% |
| 2005 | 30.50% | 7.90% | 5.10% | 4.00% | 14.80% | 37.70% |
| 2006 | 30.60% | 6.70% | 5.10% | 3.60% | 20.70% | 33.20% |
| 2007 | 31.80% | 6.60% | 5.30% | 4.10% | 18.00% | 34.30% |
| 2008 | 30.00% | 5.60% | 4.90% | 3.70% | 15.30% | 40.40% |
| 2009 | 31.30% | 5.80% | 4.20% | 3.80% | 14.50% | 40.40% |
| 2010 | 32.30% | 5.10% | 4.10% | 3.60% | 13.80% | 41.10% |
| 2011 | 31.90% | 4.70% | 4.10% | 3.90% | 13.30% | 42.20% |
| 2012 | 33.60% | 4.20% | 4.30% | 4.10% | 13.60% | 40.20% |
| 2013 | 34.70% | 4.20% | 4.30% | 4.80% | 12.50% | 39.50% |
| 2014 | 35.20% | 4.10% | 4.30% | 5.00% | 12.90% | 38.50% |
| 2015 | 37.20% | 4.10% | 4.30% | 6.00% | 12.40% | 36.00% |
| 2016 | 39.00% | 3.90% | 4.30% | 6.80% | 11.10% | 34.90% |
| 2017 | 39.20% | 3.90% | 4.60% | 7.40% | 10.70% | 34.10% |

Then, combing the export structure characteristics of disaggregated furniture with the analysis of quality values in different sub-categories, we can obtain more findings. First of all, in terms of product quality level, the three main categories all had relatively low quality, especially for wooden furniture not for kitchen, office, or bedroom use (940360), the quality of which was at the bottom. Secondly, looking at the trend of quality changes over time, we can find that the qualities of export products 940161 and 940360 decreased year by year, but the quality of product 940350 experienced a first decline and then an increase. For 940161 and 940350, the change of the export share was in the opposite direction to that of the quality. Specifically, the share of 940161 in export was rising, while its product quality was decreasing. The export share of 940350 increased first and then decreased, while the change in its product quality was just the opposite. In contrast, the proportion of 940360 in export changed in the same direction as the quality value of the product, and the export proportion and the quality value declined year after year. The decline in the quality of export products may be related to many factors, such as production productivity, R&D investment, factor costs [19], intermediate products [20], income [21], trade liberalization [22], etc., which need to be further verified by empirical tests. Among them, production productivity, R&D investment, and factor cost are the main determinants [15].

According to Equation (8), we can calculate the overall quality value of Chinese wooden furniture in exports, see Table 7. In order to identify the quality change characteristics of different categories of export wooden furniture, we made the difference between the quality of each sub-category furniture and the overall quality. The result prominently shows that Chinese wooden furniture export has entered the arduous stage of quality shrinking. It can be seen that the three major export products (940161, 940360, 940350) showed a

situation in which their qualities were lower than or basically the same level as the overall quality. The part of 940161 slightly higher than the overall quality became smaller and smaller, while the quality of 940360 was lower and lower than the overall quality. In order to break through the bottleneck that it is difficult to improve the export quality of wooden furniture, it is particularly necessary to develop and make use of the growth momentum of emerging markets to stimulate and cultivate new competitive advantages. In addition, we can see from Table 7 that the quality of wooden furniture products exported by China to the "Belt and Road" markets was significantly higher than the overall quality level, which means that a good foundation for new development has formed.

**Table 7.** Estimated quality of wooden furniture products by HS 6-digit category.

| Year | 940161 | DFO | 940169 | DFO | 940330 | DFO | 940340 | DFO | 940350 | DFO | 940360 | DFO | Overall Quality | BRI Quality |
|---|---|---|---|---|---|---|---|---|---|---|---|---|---|---|
| 1998 | 0.253 | +0.022 | 0.258 | +0.027 | 0.703 | +0.472 | 0.254 | +0.023 | 0.216 | −0.015 | 0.189 | −0.042 | 0.231 | 0.400 |
| 1999 | 0.240 | +0.017 | 0.248 | +0.025 | 0.724 | +0.501 | 0.277 | +0.054 | 0.191 | −0.032 | 0.186 | −0.037 | 0.223 | 0.402 |
| 2000 | 0.239 | +0.024 | 0.237 | +0.022 | 0.719 | +0.504 | 0.268 | +0.053 | 0.178 | −0.037 | 0.180 | −0.035 | 0.215 | 0.392 |
| 2001 | 0.223 | +0.019 | 0.237 | +0.033 | 0.715 | +0.511 | 0.265 | +0.061 | 0.178 | −0.026 | 0.178 | −0.026 | 0.204 | 0.395 |
| 2002 | 0.226 | +0.015 | 0.245 | +0.034 | 0.722 | +0.511 | 0.253 | +0.042 | 0.180 | −0.031 | 0.182 | −0.029 | 0.211 | 0.407 |
| 2003 | 0.226 | +0.011 | 0.252 | +0.037 | 0.736 | +0.521 | 0.242 | +0.027 | 0.196 | −0.019 | 0.172 | −0.043 | 0.215 | 0.407 |
| 2004 | 0.239 | +0.025 | 0.263 | +0.049 | 0.726 | +0.512 | 0.229 | +0.015 | 0.201 | −0.013 | 0.162 | −0.052 | 0.214 | 0.404 |
| 2005 | 0.248 | +0.005 | 0.261 | +0.018 | 0.718 | +0.475 | 0.238 | −0.005 | 0.202 | −0.041 | 0.186 | −0.057 | 0.243 | 0.404 |
| 2006 | 0.251 | +0.011 | 0.270 | +0.030 | 0.708 | +0.468 | 0.237 | −0.003 | 0.189 | −0.051 | 0.185 | −0.055 | 0.240 | 0.401 |
| 2007 | 0.256 | +0.008 | 0.277 | +0.029 | 0.696 | +0.448 | 0.244 | −0.004 | 0.200 | −0.048 | 0.192 | −0.056 | 0.248 | 0.394 |
| 2008 | 0.255 | +0.016 | 0.272 | +0.033 | 0.687 | +0.448 | 0.244 | +0.005 | 0.217 | −0.022 | 0.176 | −0.063 | 0.239 | 0.387 |
| 2009 | 0.254 | +0.011 | 0.284 | +0.041 | 0.695 | +0.452 | 0.228 | −0.015 | 0.230 | −0.013 | 0.186 | −0.057 | 0.243 | 0.393 |
| 2010 | 0.255 | +0.008 | 0.285 | +0.038 | 0.696 | +0.449 | 0.237 | −0.010 | 0.235 | −0.012 | 0.197 | −0.050 | 0.247 | 0.394 |
| 2011 | 0.252 | +0.000 | 0.289 | +0.037 | 0.697 | +0.445 | 0.237 | −0.015 | 0.240 | −0.012 | 0.210 | −0.042 | 0.252 | 0.404 |
| 2012 | 0.256 | +0.007 | 0.280 | +0.031 | 0.691 | +0.442 | 0.235 | −0.014 | 0.245 | −0.004 | 0.196 | −0.053 | 0.249 | 0.372 |
| 2013 | 0.251 | +0.005 | 0.281 | +0.035 | 0.684 | +0.438 | 0.235 | −0.011 | 0.245 | −0.001 | 0.192 | −0.054 | 0.246 | 0.369 |
| 2014 | 0.254 | +0.005 | 0.281 | +0.032 | 0.673 | +0.424 | 0.235 | −0.014 | 0.263 | +0.014 | 0.190 | −0.059 | 0.249 | 0.371 |
| 2015 | 0.247 | +0.001 | 0.281 | +0.035 | 0.665 | +0.419 | 0.256 | +0.010 | 0.271 | +0.025 | 0.180 | −0.066 | 0.246 | 0.384 |
| 2016 | 0.240 | +0.002 | 0.275 | +0.037 | 0.686 | +0.448 | 0.240 | +0.002 | 0.251 | +0.013 | 0.170 | −0.068 | 0.238 | 0.381 |
| 2017 | 0.236 | +0.000 | 0.274 | +0.038 | 0.703 | +0.467 | 0.236 | +0.000 | 0.246 | +0.010 | 0.165 | −0.071 | 0.236 | 0.372 |

Note: ▮ refers to positive; ▮ refers to negative. DFO refers to Difference from Overall.

### 4.2. BRI Distribution Characteristics of Chinese Wooden Furniture Quality in Exports

In 2013, President Xi Jinping put forward the initiative to jointly build the "Belt and Road Initiative", actively developing economic cooperation with countries along the "Belt and Road Initiative", bringing new economic development concepts and trade growth momentum. This section covers Mongolia and the ten ASEAN countries, the eighteen countries in West Asia including Iran and Iraq, etc., the eight South Asian countries including India, Pakistan, etc., the five countries in Central Asia including Kazakhstan, etc., the seven countries in the Commonwealth of Independent States including Russia and Ukraine, etc., and the sixteen countries in Central and Eastern Europe including Poland and Lithuania, etc. A total of 65 countries belong to the region of the "Belt and Road initiative".

From the perspective of China's total export of wood forest products to the "Belt and Road Initiative" countries, it showed an overall upward trend, rising from US $0.20 billion in 1998 to US $12.68 billion in 2018. The average annual growth rate was 27.18%. From 1998 to 2014, the export value increased year by year. In 2014, the export value reached its peak, reaching US $14.30 billion. The export declined slightly from 2014 to 2018. Among the "Belt and Road Initiative" countries, China's wood forest products export destinations are relatively concentrated. Singapore, Malaysia, the United Arab Emirates, India, Vietnam, Thailand, the Philippines, Iran, the Russian Federation, and Indonesia are the main export destinations for Chinese wood forest products, accounting for more than 65% of China's total wood forest product export to the "Belt and Road Initiative" countries. Among them,

wood forest products exported to Singapore accounted for the highest proportion, while there was a clear downward trend. The share dropped from 34.52% in 1998 to 8.23% in 2018, accounting for an average annual share of 13.00%. The proportion of wood forest products exported to Malaysia was second, and relatively stable, with an average annual proportion of 9.36%. In addition, the export proportion to the United Arab Emirates was also relatively high, showing an overall upward trend. The proportion increased from 4.01% in 1998 to 7.08% in 2018, with an average annual proportion of 8.21%. At the same time, the average annual export proportions to Thailand and India were 5.54% and 5.12%, respectively. The proportion to other countries was relatively low, with an average annual proportion of less than 5%.

Then, we put the focus on the quality of Chinese wooden furniture in the area of the "Belt and Road Initiative", to analyze its regional distribution characteristics. The quality of wooden furniture exported to the "Belt and Road Initiative" countries during 1998 to 2017 was relatively stable, while the quality declined slightly from 2012 to 2017 (see Figure 4). Specifically, the quality level was the highest in 2002, with a quality value of 0.407, whereas in 2013, the quality level was the lowest with a quality value of 0.369. Then, the top major markets were selected in terms of export value and conduct research by country. First of all, it analyzes the characteristics of the quality change trend, according to the three types of quality level rising, quality level falling, and quality change without a specific trend. The first type is that the quality level was rising. In terms of different countries, the qualities of wooden furniture exported to Malaysia, Thailand, Indonesia, and Israel were on the rise. Among them, the qualities of wooden furniture to Malaysia and Israel were relatively high. In particular, the quality of wooden furniture exported to Malaysia rose from 0.344 in 1998 to 0.390 in 2017, with an increase of 13.37%. The quality of wooden furniture to Israel increased from 0.270 in 1998 to 0.358 in 2017, having a significant increase of 32.59%. In contrast, the qualities of wooden furniture exported to Thailand and Indonesia were relatively low, and each year were lower than the overall quality level of "Belt and Road Initiative" countries. However, their qualities increased significantly. To be specific, the quality of wooden furniture to Thailand increased by 33.73%, and the quality of wooden furniture exported to Indonesia rose by 23.14%. The second type is that the quality level was falling. Furthermore, the qualities of wooden furniture exported to Singapore and Russia were showing a downward trend. Specifically, the quality of wooden furniture to Singapore was roughly the same as the overall quality of the "Belt and Road Initiative" countries, but with a slight decline. The quality value dropped from 0.426 in 1998 to 0.369 in 2017. In particular, the quality of wooden furniture exported to Russia fell sharply, and each year was lower than the overall quality level of the "Belt and Road Initiative" countries; moreover, the quality level was extremely low. The quality value dropped from 0.341 in 1998 to 0.201 in 2017. The third type is that the quality changed without a specific trend. In contrast, the qualities of wooden furniture exported to the United Arab Emirates, India, Vietnam, the Philippines, and Iran had no obvious trend of change. To be specific, the qualities of wooden furniture to the United Arab Emirates, Vietnam, and the Philippines fluctuated around the overall quality of the "Belt and Road Initiative" countries; however, the quality level is relatively high. On the contrary, the quality level of wooden furniture exported to India and Iran fluctuated constantly, and the quality level is low.

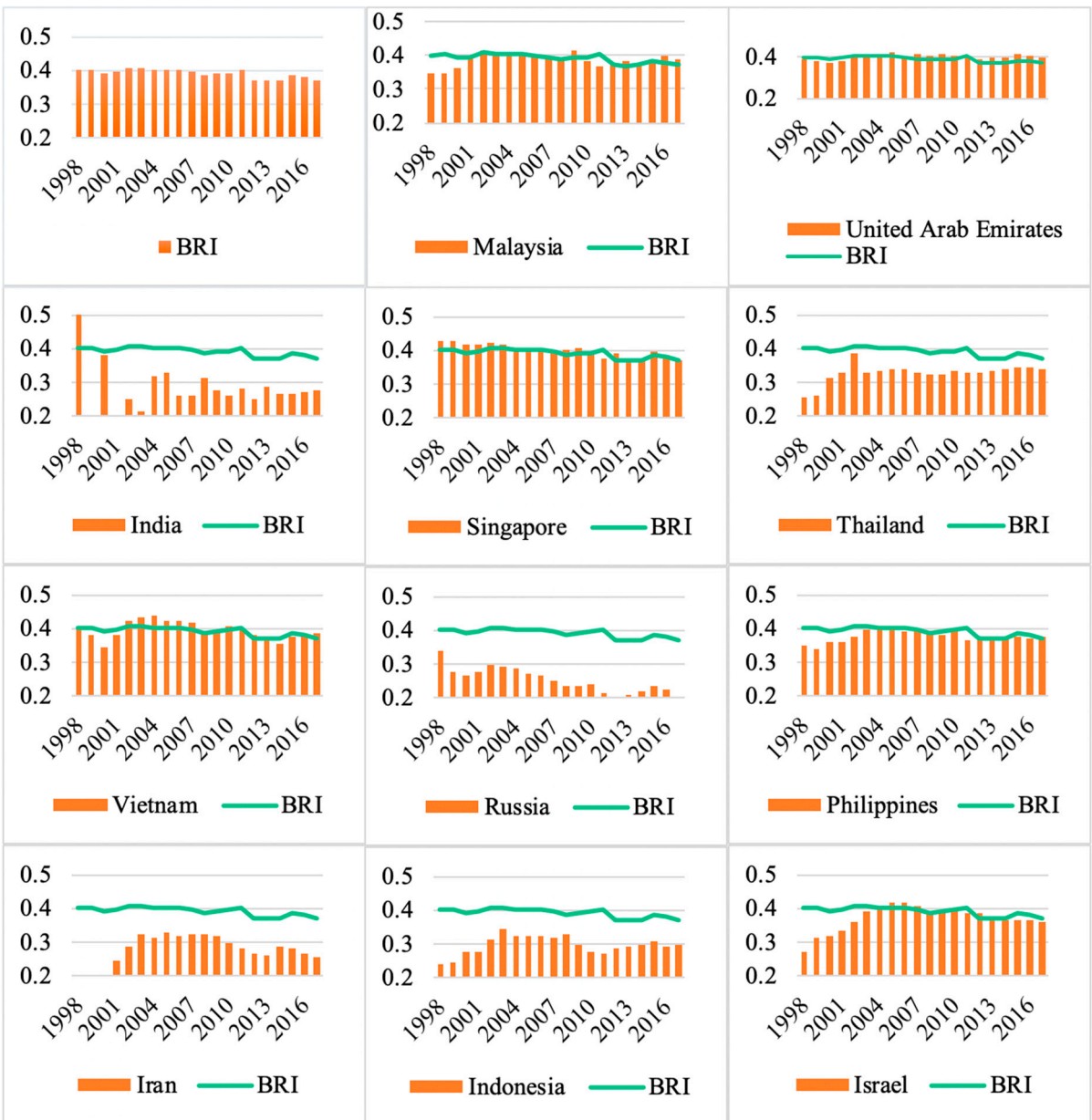

**Figure 4.** Quality of wooden furniture exported by China to major markets in the "Belt and Road Initiative" region.

## 5. Conclusions and Limitations

Based on the analysis of the export scale and competitiveness of Chinese wooden furniture, this paper uses a regression estimation method to measure the quality of wooden furniture in exports, which is the most important category in China's wood forest product exports. Furthermore, it explores the dynamic change of the quality of disaggregated wooden furniture and analyzed its distribution characteristics in the region of the "Belt and Road Initiative" countries from 1998 to 2017.

We found that: (1) Wooden furniture exports have accounted for the largest share of China's wood forest product exports for a long time, but there is a downward trend. The export competitiveness of wooden furniture is significantly stronger than other wood forest products, while it is experiencing a decline. Compared with main European competitors, the export competitiveness of Chinese wooden furniture is relatively weak; however, it shows certain competitive advantages in the Asia-Pacific region. (2) In terms of overall quality, the quality of wooden furniture in exports is lower than that of wood-based panels

and paper products, with the quality value between 0.2 and 0.25. After a slight increase in quality in 2001–2005, the quality level turns stable, but the quality level is always at a low position. The difference in its quality within the industry narrows and then expands, which conforms to the product technology development characteristics of "innovation imitation re-innovation", and is consistent with the product life cycle. Currently, each sub-category is in a new round of innovative development and the establishment of new competitive advantages. (3) From the quality level of each sub-category of Chinese wooden furniture, wooden furniture not for kitchens, offices, and bedrooms accounts for the highest proportion of exports (up to 44.34%), but the quality level is the lowest, with an average quality value of 0.184. Office wooden furniture has the highest quality level, with an average of 0.702, while it takes a relatively low percentage of total exports (only 4.26%). However, the quality of other wooden furniture is not high, between 0.2–0.3. The major three categories of wooden furniture that account for a very large share are of low quality, while the other three categories with a relatively low export value are all of higher quality, indicating that there is a strong imbalance. The quality difference of upholstered wood seats slightly higher than the overall quality is getting smaller and smaller, whereas the quality difference of wooden furniture (not for offices, kitchens, or bedrooms) is lower than the overall quality more and more. The quality of wooden furniture exported by China to "Belt and Road Initiative" markets is significantly higher than the overall level, which has formed a good foundation for development. (4) From the regional distribution characteristics of the quality in "Belt and Road Initiative" markets, the regional overall quality of Chinese wooden furniture declined slightly from 2012 to 2017. The qualities of wooden furniture exported to Malaysia, Thailand, Indonesia, and Israel show an upward trend. Specifically, the qualities of wooden furniture to Malaysia and Israel are relatively high, while that to Thailand and Indonesia are low, which is lower than the overall quality to the "Belt and Road Initiative" wooden furniture markets in each year. In contrast, the qualities of wooden furniture exported to Singapore and Russia are showing a downward trend. To be specific, the quality of wooden furniture to Singapore is roughly the same as the overall quality of the "Belt and Road Initiative" countries but with a slight decline. That to Russia has dropped significantly and remains at a low level. Those to the United Arab Emirates, Vietnam, and the Philippines are high but fluctuate up and down.

As a big trading country of wooden furniture in the world, China's share in the world market has been declining year after year. Moreover, the production is slowing down, the cost advantage is gradually weakening, technological innovation, and R&D investment are limited, which reveals that China is in a difficult period of transition from low-cost driving to high-quality driving. At the same time, Chinese furniture enterprises are facing increasingly fierce trade protection policies from developed countries such as the United States. For the furniture industry in China, where most enterprises are small and medium-sized, it is also experiencing higher and higher input costs of labor and other factors. In this case, accurately identifying the quality of different export categories of furniture products and their changing characteristics can help furniture enterprises make better production and operation decisions, promote the formation of a good business environment, and foster new comparative advantages and international competitiveness. Based on the existing research results of product quality determination and the conclusions of this paper, we propose the following suggestions in order to promote the advantages of Chinese wooden furniture exports and reduce the export disadvantages:

(1)   As the overall quality of Chinese wooden furniture in exports is still relatively low, in order to reverse this lack of high quality, based on the impact of production productivity, factor cost, and R&D investment on product quality in the endogenous quality determination model, we suggest that Chinese furniture enterprises strive to improve the quality of factor inputs, optimize, and adjust the allocation of production factors, especially increasing the proportion of technology and design inputs in production, and reducing low-skilled labor factor inputs. However, if they are small-scale furniture

enterprises, they can use the formation of industrial clusters to share technological progress and design innovation, and allocate costs to an acceptable level.

(2)  Because the quality level of Chinese wooden furniture in exports is unbalanced in different sub-categories, it is suggested that enterprises implement a product-specific quality upgrade strategy when improving the quality of wooden furniture. Specifically, for the furniture categories that account for a high share of exports but are declining in quality, that is, 940161 and 940360, as the main export sub-categories, the production of their existing products has been relatively mature. Under the current production technology, controlling factor input costs or expanding economies of scale can maintain a certain international competitive advantage, but this cost advantage strategy is not conducive to long-term competition. In particular, it will make enterprises ignore the opportunities of quality upgrading through technological innovation. Therefore, Chinese furniture enterprises should focus on increasing the innovation investment in those mature products and cultivating new competitive advantages. In addition, the entire furniture industry can collaboratively improve the use of new technologies, new materials, and the development of new products, while at the same time strengthen the protection of intellectual property rights, creating an industry motivation for quality improvement.

(3)  It is suggested that Chinese furniture enterprises implement differentiated regional development strategies for different markets. For traditional wooden furniture markets, such as the United States and the European Union, enterprises should actively participate in fair market competition by exporting more competitive wooden furniture categories, so as to realize the survival of the fittest. On the one hand, by using the "learning by doing" effect of exports, the quality of wooden furniture can be improved by increasing the productivity of enterprises. On the other hand, for capable enterprises, they can obtain higher design and innovation capabilities through investment or acquisition of the furniture enterprises in the traditional market, and more easily integrate into the local market. For emerging markets such as the BRI countries, considering the differentiation characteristics of Chinese wooden furniture quality changes to different destinations, it is recommended that furniture enterprises take advantage of the market where the quality of Chinese wooden furniture is on the rise, to strengthen regional and bilateral trade cooperation to further expand the export value of specific products. However, for the BRI markets where the quality of export products is changing downwards, it is suggested that furniture enterprises promote the low-proportion and high-quality disaggregated wooden furniture categories advancing to more destination countries with demand.

On the basis of this study, the article can be extended in the following area: export product quality measures. This paper adopts the quality measure and ShiBing exhibition (2014), Xu and Even (2016). Consistent with the measurement methods, there are a number of export product quality measurement methods. Each measurement method has its advantages and characteristics. In the follow-up study, multiple methods can be used to measure and verify each other to enhance the robustness and reliability of the conclusions.

**Author Contributions:** Conceptualization, L.W., N.B. and Y.F.; methodology, L.W., N.B., Y.F. and L.Y.; formal analysis, L.W., N.B., Y.F. and L.Y.; investigation, N.B. and L.Y.; resources, L.W.; writing—original draft preparation, L.W., N.B., Y.F. and L.Y.; writing—review and editing, L.W. and N.B.; funding acquisition, L.W. All authors have read and agreed to the published version of the manuscript.

**Funding:** This research was funded by Beijing Social Science Fund grant number 20JJC026 and The APC was funded by Beijing Forestry University.

**Data Availability Statement:** Not applicable.

**Conflicts of Interest:** The authors declare no conflict of interest.

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
