# Peer review of "Product Quality Measurement, Dynamic Changes, and the Belt and Road Initiative Distribution Characteristics: Evidence from Chinese Wooden Furniture Exports"

_forests, doi:10.3390/f13071153_

Round 1

Reviewer 1 Report

The paper brings results of the study focused on the quality of export in Chinese furniture products. The focus of the paper suits a scope of the journal.

The aim of the paper is unclear. The structure of the paper could be improved towards a structure of a scientific paper.

My comments for improvement of the paper are as follows:

1.      Abstract and introduction: state the aim of the paper from the scientific point of view clearly

2.      I am not sure if Product quality measurement in the title of the paper is corresponding with the paper´s content – please revise

3.      Keep the structure of a scientific paper: introduction, (literature review), materials and methods, results, discussion, conclusions. Now, there is a mix of methodology and results.

4.      Discussion: discuss the results of own research with the results of other researchers – add the information to analysis of results.

5.      Conclusion: Give limitations of provided research and directions of the future research.

Author Response

Thanks a lot for your comments and appreciate your help!

Response to Reviewer 1 Comments

Point 1: The aim of the paper is unclear.

Response 1: Revised the abstract, discussion and conclusion. In summary, this paper accurately identifies the quality of different export categories of furniture products and their changing characteristics which can help furniture enterprises make better production and operation decisions, as well as promoting the formation of a good business environment and fostering new comparative advantages and international competitiveness. It has laid a good foundation for Chinese furniture companies to explore and utilize emerging markets.

Point 2: The structure of the paper could be improved towards a structure of a scientific paper.

Response 2: Revised and restructured as scientific paper with introduction, methodology, results, discussion and conclusion.

Point 3: Abstract and introduction: state the aim of the paper from the scientific point of view clearly

Response 3: Done.

Point 4: I am not sure if Product quality measurement in the title of the paper is corresponding with the paper´s content – please revise--argue

Response 4: Product quality measurement(and dynamic change) is the key word in this paper. In this case, this paper aims to accurately identify the quality of different export categories of furniture products and their changing characteristics using the Belt and Road country data sample. The outcome is to help our furniture enterprises make better production and operation decisions, promote the formation of a good business environment, and foster new comparative advantages and international competitiveness.

Point 5: Keep the structure of a scientific paper: introduction, (literature review), materials and methods, results, discussion, conclusions. Now, there is a mix of methodology and results.

Response 5: Done.

Point 6: Discussion: discuss the results of own research with the results of other researchers – add the information to analysis of results.

Response 6: Added to the discussion part.

Reviewer 2 Report

This is an interesting paper and the paper provide a interesting perspective on the Chinese wooden furniture export and its product quality measurement, dynamic changes and the Belt and Road initiative distribution characteristics.

The paper is generally well-written. However, I am not sure whether this is a research article or a review paper, as it was not mentioned in the paper. If this a research article, please prepare the manuscript in a structure of a research article: Introduction, methodology, results, discussion and conclusion.

Apart from that, I found that most of the statement is lack of references, or without sources. Is it a fact statement or it’s just a speculation from the authors? Please fix this issue.

Section 1 – OK

Section 2.1 – can you please revise the subheading? It’s too long

Can you please make a table for the definition of wood forest products between Chinese research institutions and FAO so that the readers can easily spot out the difference?

Table 1 was not mention in the text. The title of the Table should be at the top of the Table, not bottom of the Table.

Please add source for Table 1.

Figure 1 – please mention clearly that the percentage of the pie chart in the middle of Figure 1 represents an average export percentage between 1998 to 2018. Readers will get confused easily.

Figure 1 – “The proportion of the export value of wood forest products” of what? China? Worldwide? Please mention clearly.

Section 2.2 - can you please revise the subheading? It’s too long

How was the revealed advantage index (RCA index) calculated? On what basis that RCA>2.5 represents strong competitiveness which lower than 0.8 means weak competitiveness? Any reference?

Figure 2 – x-axis - 1988.00, 1999.00…. please reduce the decimal point.

Section 2.3 - can you please revise the subheading? It’s too long

Table 2 is a bit confusing, the coloured circle is too close to the next value. At first glance, it thought 1.65 is represent by red circle (moderate), but actually it is strong. Please fix it.

Section 3 – nicely done, well written

Section 4 – well written.

Author Response

Thanks a lot for your comments and appreciate your help!

Response to Reviewer 2 Comments

Point 1: However, I am not sure whether this is a research article or a review paper, as it was not mentioned in the paper. If this a research article, please prepare the manuscript in a structure of a research article: Introduction, methodology, results, discussion and conclusion.

Response 1: Rewrite the manuscript in a structure of a research article: introduction, methodology, results, discussion and conclusion.

Point 2: Apart from that, I found that most of the statement is lack of references, or without sources. Is it a fact statement or it’s just a speculation from the authors? Please fix this issue.

Response 2: Some are quoted with footnotes, some are quoted in text, some are fact statement like the Belt and Road idea and data, some are speculations from authors as this is with some subjective forecasting.

Point 3: Section 2.1 Can you please revise the subheading? It’s too long

Response 3: Done.

Point 4: Can you please make a table for the definition of wood forest products between Chinese research institutions and FAO so that the readers can easily spot out the difference?

Response 4: Make a new table, please refer to new Table 1 in manuscript.

Point 5: Table 1 was not mention in the text.The title of the Table should be at the top of the Table, not bottom of the Table.Please add source for Table 1.

Response 5: Table 1 is used for table 4 explanation, source added and title put at the top.

Point 6: Figure 1 – please mention clearly that the percentage of the pie chart in the middle of Figure 1 represents an average export percentage between 1998 to 2018. Readers will get confused easily.

Response 6: Done.

Point 7: Figure 1 – “The proportion of the export value of wood forest products” of what? China? Worldwide? Please mention clearly.

Response 7: Done.

Point 8: Section 2.2 - can you please revise the subheading? It’s too long

Response 8: Done.

Point 9: How was the revealed advantage index (RCA index) calculated? On what basis that RCA>2.5 represents strong competitiveness which lower than 0.8 means weak competitiveness? Any reference?

Response 9: Done and made clear under the table.

Point 10: Figure 2 – x-axis - 1988.00, 1999.00…. please reduce the decimal point.

Response 10: Done.

Point 11: Section 2.3 - can you please revise the subheading? It’s too long

Response 11: Done.

Point 12: Table 2 is a bit confusing, the coloured circle is too close to the next value. At first glance, it thought 1.65 is represent by red circle (moderate), but actually it is strong. Please fix it.

Response 12: Done.

Please see the updated version of manuscript in the attachment.

Round 2

Reviewer 1 Report

The paper was revised according to comments sufficiently.